# Self-Hybridized Exciton-Polaritons in Sub-10-nm-Thick WS_2_ Flakes: Roles of Optical Phase Shifts at WS_2_/Au Interfaces

**DOI:** 10.3390/nano12142388

**Published:** 2022-07-13

**Authors:** Anh Thi Nguyen, Soyeong Kwon, Jungeun Song, Eunseo Cho, Hyohyeon Kim, Dong-Wook Kim

**Affiliations:** Department of Physics, Ewha Womans University, Seoul 03760, Korea; nthianh111@gmail.com (A.T.N.); kwonsso91@gmail.com (S.K.); sje10056996@gmail.com (J.S.); escho797@gmail.com (E.C.); kimhyohyun1614@gmail.com (H.K.)

**Keywords:** WS_2_, exciton–polariton, anticrossing, Fresnel coefficients, phase shift

## Abstract

Exciton–polaritons (EPs) can be formed in transition metal dichalcogenide (TMD) multilayers sustaining optical resonance modes without any external cavity. The self-hybridized EP modes are expected to depend on the TMD thickness, which directly determines the resonance wavelength. Exfoliated WS_2_ flakes were prepared on SiO_2_/Si substrates and template-stripped ultraflat Au layers, and the thickness dependence of their EP modes was compared. For WS_2_ flakes on SiO_2_/Si, the minimum flake thickness to exhibit exciton–photon anticrossing was larger than 40 nm. However, for WS_2_ flakes on Au, EP mode splitting appeared in flakes thinner than 10 nm. Analytical and numerical calculations were performed to explain the distinct thickness-dependence. The phase shifts of light at the WS_2_/Au interface, originating from the complex Fresnel coefficients, were as large as π/2 at visible wavelengths. Such exceptionally large phase shifts allowed the optical resonance and resulting EP modes in the sub-10-nm-thick WS_2_ flakes. This work helps us to propose novel optoelectronic devices based on the intriguing exciton physics of TMDs.

## 1. Introduction

Transition metal dichalcogenides (TMDs) have long attracted the attention of researchers due to their fascinating physical properties for optoelectronic device applications, including sizable bandgap energies, large absorption coefficients, high electron mobility, and superior mechanical flexibility [1,2,3,4,5,6,7,8,9,10,11,12,13,14,15]. In particular, it has been noted that the exciton–photon interactions dominate the optical responses of TMDs even at room temperature. In two-dimensional (2D) TMD layers, strongly bound electron-hole pairs (excitons) are generated by the weak dielectric screening and strong geometric confinement [1,2,3,4,5,6,7,8,9,10,11]. The exciton binding energy in TMDs is as large as hundreds of meV, which is one to two orders of magnitude larger than those of conventional semiconductors [1,2,3,4,5,6]. Integration of TMDs with photonic nanostructures can broaden our understanding of exciton physics, providing valuable insights into excitonic devices [4,5,6]. Moreover, strong coupling between excitons and photons leads to the formation of exciton–polaritons (EPs) in TMDs integrated with optical resonators [5,6]. These half-light half-matter quasiparticles allow us to investigate intriguing physical phenomena [16,17,18] and realize novel functional devices [19,20,21].

TMD multilayer flakes, with exceptionally large refractive indices, can sustain Fabry–Pérot (FP) resonance modes without any external cavity [8,9,10,11,12]. Consequently, a variation in the thickness significantly alters the apparent color of the TMD flakes due to the absorption and interference effects. Such unique optical characteristics enable the rapid and reliable determination of the TMD thicknesses using optical microscopy (OM) [13]. Moreover, coupling of excitons and the cavity photons in TMDs results in the formation of self-hybridized EPs [8,9,10,11,12]. Consequently, exciton–photon anticrossing behaviors appear at specific wavelengths along with splitting of the hybridized EP modes to upper and lower polariton branches (UPB and LPB, respectively) in TMD multilayer flakes on reflective substrates [6,8,9,10] The wavelength showing such cavity-free EP mode splitting depends on the thickness of the TMD flakes, since the flake thickness directly determines the optical resonance wavelength [8,9,10]. The integration of TMDs with optical resonators requires complicated time-consuming fabrication processes which are obstacles for active research works and the development of excitonic devices. Therefore, cavity-free self-hybridized EPs provide a versatile approach to investigate excitonic effects in the optical characteristics of TMDs.

Among numerous TMDs, WS_2_ is one of the most intensively investigated materials. Monolayer WS_2_ has a direct bandgap of ~2 eV, while the bulk counterpart has an indirect bandgap of ~1.3 eV [22,23,24]. Since the bandgap is appropriate for visible-range applications, WS_2_ is a strong candidate to realize high-performance optoelectronic devices [23]. Additionally, WS_2_ is a promising material for valleytronic devices, due to its broken inversion symmetry and strong spin–orbit coupling [24]. Earlier reports have shown that the minimum thickness of a WS_2_ flake on dielectric SiO_2_ substrates showing UPB-LPB splitting is approximately 40 nm, which is less than ~1/10 of the exciton resonance wavelength in vacuum [8]. To our surprise, exciton–photon anticrossing behaviors appear even in sub-20-nm-thick TMDs on metallic layers [10]. The EP mode formation boosts the optical absorption in thin TMD layers, which has stimulated the development of high-efficiency ultrathin photovoltaic devices [14,15]. Despite these noteworthy features, the physical origin to determine the minimum thickness for EP mode splitting in TMDs has not been explicitly investigated.

In this work, we prepared exfoliated WS_2_ multilayer flakes and investigated their optical characteristics. The apparent colors and the measured reflectance spectra of the flakes showed significant variation depending on the flake thickness, which originated from the strong coupling between excitons and cavity photons. In particular, the thickness-dependent spectral responses of the flakes on SiO_2_/Si substrates and Au thin films were compared to study how the metal underlayers affected the optical resonance and resulting EP mode splitting. Analytical and numerical calculations were also performed to elucidate the physical origins.

## 2. Materials and Methods

Exfoliated WS_2_ flakes were prepared on SiO_2_ (300 nm)/Si wafers and Au (100 nm) thin films, as illustrated in the schematic diagrams in Figure 1a,b. Hereafter, the former and latter samples will be called WS_2_/SiO_2_/Si and WS_2_/Au, respectively. The Au thin films were deposited on SiO_2_/Si substrates using e-beam evaporation and then were delaminated from the original substrates to slide glasses using UV-curable prepolymer (NOA63, Norland) [25]. The SiO_2_/Si substrates serve as ultrasmooth templates, and the stripped Au thin films have very flat surface (typical root-mean-square roughness: 0.7 nm) (see Appendix A). Such template-stripped Au thin films are beneficial for minimizing the roughness at the interface between the WS_2_ flakes and Au thin films [25].

Optical reflectance spectra of flakes were measured using a homemade setup with an optical microscope (LV100, Nikon, Tokyo, Japan) and a spectrometer (Maya 2000 Pro, Ocean Optics, Dunedin, FL, USA). Reflected light from sample surface was collected using a 50-μm-diameter optical fiber (M50L02S-A, Thorlabs, Newton, MA, USA), which enabled us to obtain the spectra from a selected area of several μm^2^. The thickness of the flake was measured using an atomic force microscopy system (NX10, Park Systems, Suwon, Korea).

## 3. Results and Discussions

As shown in Figure 1a,b, the thickness of the WS_2_ flake (*d*_WS2_) significantly affects the apparent color of the flakes. It should be also noted that the flakes with identical *d*_WS2_ values (e.g., 4, 7, and 20 nm) exhibit distinct colors depending on the underlying layers (SiO_2_/Si and Au). Incident light undergoes reflection and transmission at the interface of two neighboring media in WS_2_/SiO_2_/Si and WS_2_/Au. The amplitude and phase of light at each medium are determined by the Fresnel equations [13,26]. The superposition of all the reflected waves at the boundaries, i.e., the multiple-beam interference, determines the reflectance spectra and colors of the WS_2_ flakes [26]. Even though the absorption coefficient of WS_2_ is exceptionally large compared with those of conventional semiconductors, thin flakes allow the transmission of incident light [23]. As a result, the underlying layer as well as *d*_WS2_ can affect the apparent color of the flake, as shown in Figure 1a,b.

Figure 2a–e shows the calculated and measured reflectance spectra of WS_2_ flakes. The calculated spectra were obtained using transfer matrix method (TMM) based on the refractive indices of the materials in the literature [27,28]. The spectra of stand-alone WS_2_ flakes are also calculated for comparison with those of WS_2_/SiO_2_/Si and WS_2_/Au (Figure 2a). All the spectra exhibit local minima at 620 and 510 nm, which correspond to the A and B exciton resonance wavelengths of multilayer WS_2_, respectively (see dashed lines in Figure 2a–e) [8,9,10,23] WS_2_ flakes with *d*_WS2_ > 40 nm exhibit thickness-dependent reflectance dips in addition to the exciton resonance dips (Figure 2a–c). These thickness-dependent reflectance dips originate from EP mode splitting since WS_2_ multilayer flakes can work as optical cavities without external cavities [8,9,10]. The EPs can be formed when the wavelengths of the exciton resonances and cavity modes are close to each other. For an intuitive understanding, FP cavity modes in dielectric thin films can be regarded as standing waves. Thus, the minimum *d*_WS2_ for the FP cavity mode is expected to be either *λ*/2*n*_WS2_ or *λ*/4*n*_WS2_ (*n*: real part refractive index of WS_2_) [26]. Considering the large refractive index of WS_2_, the minimum *d*_WS2_ allowing the FP resonance can be less than 100 nm. Wang et al. reported the EP mode splitting from WS_2_ flakes with *d*_WS2_ > 40 nm on glasses [8]. Our results also show that the exciton–photon anticrossing occurs in sub-100-nm-thick WS_2_ flakes (Figure 2a–c). Interestingly, the EP-mediated reflectance dips appear in WS_2_/Au even with *d*_WS2_ < 40 nm (Figure 2c). Zhang et al. reported similar thickness-dependent reflectance spectra from WS_2_ multilayer flakes on Au thin films [10].

Figure 2d,e shows the measured reflectance spectra of WS_2_/SiO_2_/Si and WS_2_/Au with several *d*_WS2_ values, respectively. These experimental data well reproduce the key features of the calculation results in Figure 2b,c. The samples with *d*_WS2_ of 50~60 nm show two EP-induced reflectance dips in both WS_2_/SiO_2_/Si and WS_2_/Au, similar to others’ reports. [8,9,10] Hereafter, the dips at the wavelength (*λ*) < 600 nm and *λ* > 600 nm will be called as UPB- and LPB-related dips, respectively (see gray circles in Figure 2b–e). The UPB- and LPB-related reflectance dips also appear in the stand-alone flakes (Figure 2a). This suggests that the EP mode splitting observed in the flakes with *d*_WS2_ of several tens of nm is originated from the self-hybridization of the excitons and the FP cavity photons [8,9,10] The reflectance spectra of WS_2_/SiO_2_/Si with *d*_WS2_ = 8, 20, and 27 nm exhibit exciton resonance-mediated dips (see dashed lines in Figure 2d). These spectra show a gradual increase in the reflectance at long wavelengths above 700 nm, which depends on the thickness of the SiO_2_ layer (see Appendix A). Thus, these long-wavelength features can be attributed to thin film interference in 300-nm SiO_2_/Si substrates. In WS_2_/Au with *d*_WS2_ of 10 nm, a broad UPB-induced dip appears at 500 nm < *λ* < 600 nm and a LPB-induced dip appears slightly above the A exciton resonance wavelength (see gray triangles in Figure 2c,e). WS_2_/Au with *d*_WS2_ of 21 and 25 nm shows very broad UPB-mediated dips at *λ*~600 nm and weak LPB-related dips at *λ* > 600 nm (see gray triangles in Figure 2c,e). These measured reflectance dips of WS_2_/Au agreed well with the calculation data (Figure 2c). This suggests that self-hybridized EP mode splitting can be formed in WS_2_ flakes with *d*_WS2_ much smaller than *λ*/4*n*_WS2_.

Figure 3a,b shows that the measured and TMM-calculated reflectance spectra of WS_2_/Au with *d*_WS2_ ≤ 30 nm well agreed with each other. This suggests that the samples were well prepared, as intended. In particular, the good agreement between the experimental and calculated results should be attributed to the ultrasmooth surface of the template-stripped Au thin films. In the calculations, the refractive indices of the bulk WS_2_ in Ref. [28] were used, since most of the flakes considered in this work were thicker than several layers. The UPB-related dips exhibit red-shift as increasing *d*_WS2_ and they merge with the A-exciton dips. The LPB-related dips appear near the A exciton wavelength and exhibit red-shift as increasing *d*_WS2_. From the measured and calculated data, the Rabi splitting energy of WS_2_ flakes with *d*_WS2_ < 50 nm on Au was estimated to be ~180 meV, which is somewhat smaller than those of flakes with *d*_WS2_ > 50 nm reported in literature: ~270 meV in Ref. [8] and ~235 meV in Ref. [9] (see Appendix A). Figure 2c and Figure 3a,b clearly show that the EP mode dips can appear in even sub-10-nm-thick WS_2_ flakes on Au. Such thickness is much smaller than *λ*/4*n*_WS2_. Therefore, the simple analogy between the FP cavities and the standing waves in air columns cannot satisfactorily explain the minimum *d*_WS2_ forming the EP modes in WS_2_ flakes.

The electric field (E-field) distributions in WS_2_/SiO_2_/Si and WS_2_/Au can be obtained using finite-difference time-domain (FDTD) simulation, as shown in Figure 4a–d. The horizontal axis, *z*, represents the position along the direction perpendicular to the sample surface. The origins of *z* were chosen at the WS_2_/SiO_2_ and WS_2_/Au interfaces for WS_2_/SiO_2_/Si and WS_2_/Au, respectively. The regions between the dashed lines represent the WS_2_ flakes with *d*_WS2_ = 70 nm for Figure 4a,b and *d*_WS2_ = 20 nm for Figure 4c,d. The left sides of the WS_2_ regions indicate the underlying SiO_2_ (300 nm) and Au (100 nm) layers for WS_2_/SiO_2_/Si (Figure 4a,c) and WS_2_/Au (Figure 4b,d), respectively. The right sides of the WS_2_ flakes indicate air. The E-field distributions were obtained at wavelengths where the local minima of the reflectance appear (Figure 2b,c). The field distribution for WS_2_(70 nm)/SiO_2_/Si at *λ* = 580 nm, corresponding to UPB, is similar to the fundamental-mode standing wave in a pipe opened at both ends (Figure 4a). At *λ* = 650 nm, corresponding to LPB, the WS_2_/SiO_2_ interface looks like the antinode of a standing wave, but the other antinode is in air rather than the WS_2_ surface. The field distribution for WS_2_/Au can be compared to the standing wave pattern of a one-side-closed air column. In the cases of WS_2_(70 nm)/Au at *λ* = 555 (UPB) and 630 nm (LPB), the WS_2_/Au interface is similar to the node of a standing wave, but the antinode is in air (Figure 4b). The E-field exponentially decays in the Au layer due to the absorption and the decay length is determined by the penetration depth of light in Au (for example, 50 nm at *λ* = 600 nm) [29].

The *d*_WS2_ of 20 nm is only ~1/10 of the wavelength of visible light, even if the large refractive index of WS_2_ (4~5) is considered. Thus, the magnitude of the E-field does not change much in the 20-nm-thick WS_2_ flakes (Figure 4c,d). As shown in Figure 4c, a large E-field appears in the 20-nm-thick WS_2_ flake on SiO_2_/Si under 600-nm-wavelength light illumination. Since the absorption is proportional to the square of the E-field, the large E-field increases (reduces) the absorption in the flake (reflectance), as shown in Figure 2b,d. In WS_2_(20 nm)/Au, the local minima in the reflectance appear at *λ* = 600 and 700 nm (Figure 2c). The magnitude of the E-field in the flake at *λ* = 600 nm is smaller than that at *λ* = 700 nm (Figure 4d). Since the optical absorption depends on the imaginary part of the permittivity of the medium as well as the E-field, the smaller reflectance of WS_2_ (20 nm)/Au at *λ* = 600 nm seems to result from the very large absorption coefficient of WS_2_ near the exciton resonance wavelength. At *λ* = 700 nm (LPB), the relatively large E-field increases the absorption in the WS_2_ flake, resulting in a broad reflectance dip at *λ* = 700 nm (Figure 2c). Obviously, the cavity modes in WS_2_/Au with *d*_WS2_ < 40 nm are distinct from typical FP resonance modes, which can be regarded as standing waves.

As illustrated in Figure 5a, three kinds of optical phase shifts need to be considered to understand the reflectance spectra of WS_2_/SiO_2_/Si and WS_2_/Au. The spectra of the multilayers can be explained by superposition of light reflected at each interface, i.e., multiple-beam interference [26]. First, the propagation-related phase shift of light, *φ*_WS2_, is given by 22πλnWS2dWS2, as shown in Figure 5b. Clearly, *φ*_WS2_ increases with increasing *d*_WS2_. Notably, *φ*_WS2_ of a 50-nm-thick WS_2_ flake becomes larger than 2π. *n*_WS2_ is as large as 4~5 in the visible wavelength range and, therefore, the propagation of light in the WS_2_ flake with a thickness of only about ~1/10 of the vacuum wavelength results in a large *φ*_WS2_ [23]. The Fresnel coefficient of the reflected light, *r_ij_*, is $\frac{n_{i}-n_{j}}{n_{i}+n_{j}}$ (*n_i_* and *n_j_* are the refractive indices of the two media at the interface) [26]. If *n_i_* and *n_j_* are real, then *r_ij_*, is also real and the phase of *r_ij_* can be either 0 (*r_ij_* ≥ 0) or π (*r_ij_* < 0). However, the phase of *r_ij_* can differ from either 0 or π for complex *n_i_* and *n_j_*. The Fresnel coefficient-related phase shifts at the WS_2_/air (*φ*_Air_), WS_2_/SiO_2_ (*φ*_SiO2_), and WS_2_/Au (*φ*_Au_) interfaces are shown in Figure 5c. The optical phase shifts largely vary as a function of wavelength, depending on the complex refractive indices of WS_2_ and Au. The magnitude of *φ*_Au_ is much larger than that of *φ*_Air_ and *φ*_SiO2_ over a broad wavelength range since the complex refractive index of Au is very large [27,29].

Figure 5d shows the total round-trip phase shift of light in WS_2_/Au, *φ*_Total_, which is the sum of the interfacial Fresnel coefficient-related contributions (*φ*_Air_ and *φ*_Au_) as well as the propagation-related change (*φ*_WS2_). When *φ*_Total_ is equal to integer multiples of 2π, WS_2_/Au can form resonant cavity modes. Such resonant modes boost the optical absorption in the WS_2_ flakes, as featured as local minima in the optical reflectance spectra. The intersection points of the *φ*_Total_ and 2π(integer) curves represent the resonant cavity mode wavelengths for WS_2_/Au. The intersection points in Figure 5d agree well with the local minima in the calculated and measured spectra in Figure 2c,e. For *d*_WS2_ > 40 nm, *φ*_WS2_ is larger than 2π at a certain wavelength, giving rise to the FP resonance modes and resulting EP-induced reflectance dips. Figure 5d also shows that the resonant modes can appear for *d*_WS2_ < 30 nm. For example, WS_2_/Au of *d*_WS2_ = 10 nm possesses intersection points near *λ* = 600 nm, close to the A exciton resonance wavelength of WS_2_. As a result, the reflectance spectra of sub-10-nm-thick WS_2_ flakes on Au exhibit clear features of UPB and LPB modes, as shown in Figure 2c and Figure 3a,b. It should be noted that *φ*_WS2_ alone cannot enable the resonant cavity mode in WS_2_/Au of *d*_WS2_ < 10 nm (Figure 5b). Since the negative *φ*_Au_ can reduce *φ*_Total_ to zero at certain wavelengths, WS_2_/Au enables the coupling of excitons and cavity photons (Figure 5c,d). The considerable contribution of the interface phase shift can explain why the E-field distributions in WS_2_/Au are distinct from the waveforms of simple standing waves (Figure 4b,d).

Figure 6a,b shows the measured reflectance spectra of WS_2_ flakes on SiO_2_/Si substrates with *d*_WS2_ < 30 nm before and after deposition of 30-nm-thick Au thin films, respectively. The top Au layers significantly modify the reflectance spectra of the WS_2_ flakes. Such very thin flakes without Au thin films cannot exhibit the EP mode splitting, and there are only exciton-related dips in the reflectance spectra (Figure 2b and Figure 6a). However, the Au-coated flake exhibits additional reflectance dips in addition to the exciton-related dips, indicating the self-hybridized EP modes (Figure 6b). Since the penetration depth of Au at *λ* = 600 nm is 50 nm, the 30-nm-thick Au thin films allow the transmission of incident light to the WS_2_ flakes and the formation of optical cavity modes [29]. The TMM-calculated 3D reflectance plots of Au(30 nm)/WS_2_/SiO_2_/Si samples in Figure 6c clearly show the exciton–photon anticrossing behaviors for not only *d*_WS2_ > 40 nm but also *d*_WS2_ < 40 nm, similar to WS_2_/Au (Figure 2c and Figure 3a,b). The additional dips in the measured spectra (Figure 6b) agree well with those in the calculations (Figure 6c). These experimental and calculated results suggest that the contribution of the large optical phase shifts at the Au/WS_2_ interface is crucial for the formation of EPs in WS_2_ flakes with *d*_WS2_ < *λ*/4*n*_WS2_. Many TMDs possess large real and imaginary parts of refractive indices [11,12,13,14,15,30], and hence the EP modes are expected in other kinds of very thin TMD layers. Furthermore, strong electronic interactions [31] as well as remarkable plasmonic effects [7,10] at TMD/metal interfaces can give rise to emergent physical phenomena of TMD/metal systems.

## 4. Conclusions

We investigated the reflectance spectra of exfoliated WS_2_ multilayer flakes on SiO_2_/Si substrates and template-stripped Au layers. On such reflective layers, the exceptionally large refractive indices of WS_2_ gave rise to optical resonance modes in the flakes without external cavities. The reflectance spectra of the flakes exhibited not only exciton-resonance-mediated dips but also EP mode-induced dips, resulting from hybridization of the excitons and the cavity photons. EP mode splitting appeared in WS_2_/SiO_2_/Si with *d*_WS2_ > 40 nm, whereas WS_2_/Au, even with *d*_WS2_ < 10 nm, exhibited EP mode splitting. Such a notable difference in the minimum thickness for the formation of EPs could be attributed to the large optical phase shifts at the WS_2_/Au interface. These results suggested that integration of metal thin films and nanostructures with TMDs enabled control of the EP behaviors and resulting optical characteristics of the TMD/metal systems.

## Figures and Tables

**Figure 1 nanomaterials-12-02388-f001:**
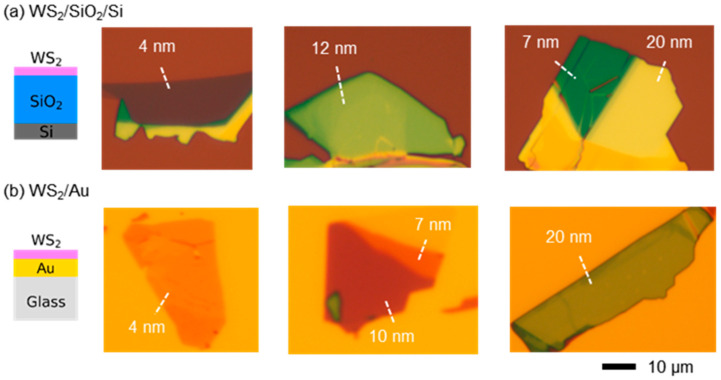
Cross-sectional schematic diagrams and OM images of exfoliated WS_2_ flakes with various thicknesses on (**a**) SiO_2_/Si substrates and (**b**) template-stripped Au films.

**Figure 2 nanomaterials-12-02388-f002:**
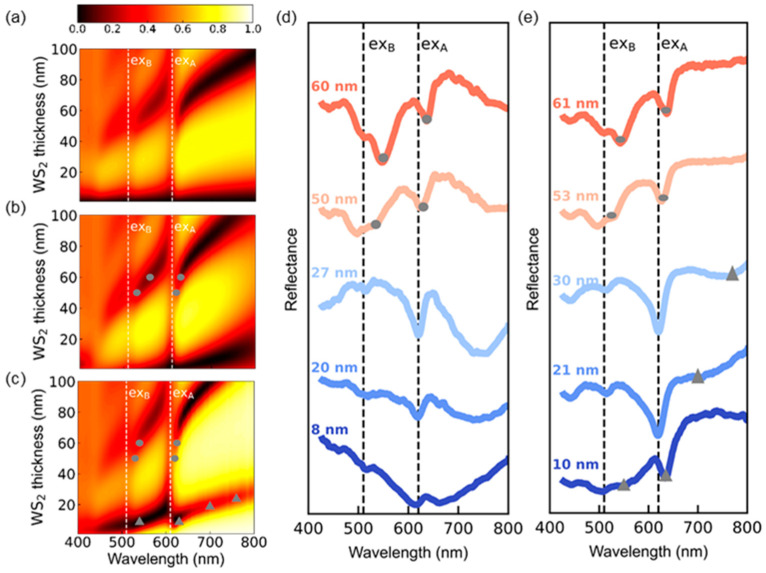
(**a**) TMM-calculated thickness-dependent reflectance spectra of (**a**) stand-alone WS_2_ (air/WS_2_/air), (**b**) WS_2_/SiO_2_/Si, and (**c**) WS_2_/Au. Measured reflectance spectra of WS_2_ flakes with various thicknesses on (**d**) SiO_2_/Si substrates and (**e**) Au thin films. The exciton resonance wavelengths of WS_2_ are denoted as dashed lines in a–e. Gray circles and triangles in b–e represent the EP-induced dips.

**Figure 3 nanomaterials-12-02388-f003:**
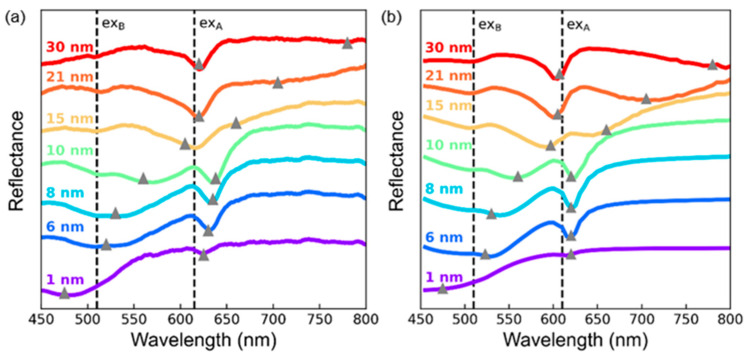
(**a**) Measured and (**b**) TMM-calculated thickness-dependent reflectance spectra of WS_2_ flakes on the Au films. The dashed lines indicate the exciton resonance wavelengths of WS_2_ multilayers. The gray triangles represent the EP-induced reflectance dips.

**Figure 4 nanomaterials-12-02388-f004:**
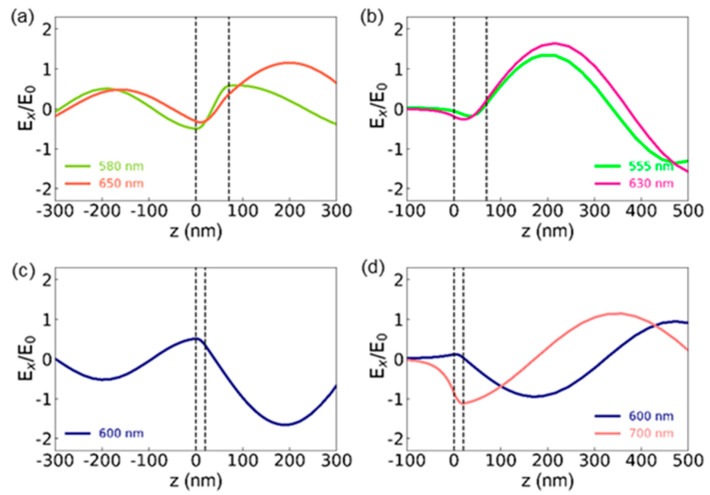
FDTD-calculated E-field distributions of 70-nm-thick WS_2_ flakes on (**a**) SiO_2_/Si and (**b**) Au and 20-nm-thick WS_2_ flakes on (**c**) SiO_2_/Si and (**d**) Au. The wavelengths correspond to the local minima of the reflectance spectra (Figure 2b,c). The regions between two dashed lines represent the WS_2_ flakes. The left and right sides of the dashed lines correspond to the underlying layers (SiO_2_ and Au) and air, respectively. The z-axis is perpendicular to the sample surface and the origin is set at the WS_2_/substrate interface. *E*_0_ indicates the magnitude of the E-field of the incident light.

**Figure 5 nanomaterials-12-02388-f005:**
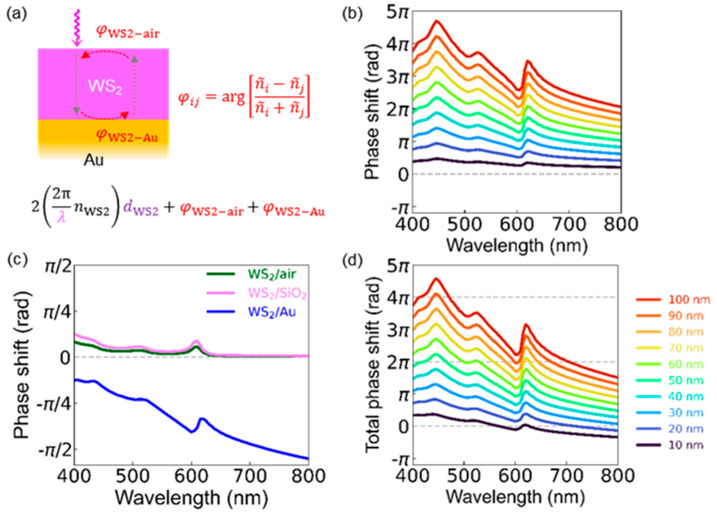
(**a**) Schematic diagram to illustrate the phase shift of light in WS_2_/Au (detailed explanations can be found in the text). (**b**) Calculated wavelength-dependent phase shift caused by round-trip propagration in WS_2_ flakes with a *d*_WS2_ of 10~100 nm. (**c**) Complex Fresnel coefficient-related phase shifts of light reflected at the WS_2_/air, WS_2_/SiO_2_, and WS_2_/Au interfaces. (**d**) Total phase shifts of light for WS_2_/Au as a function of wavelength. The intersection points of the phase shift curves and dashed lines (0, 2π, and 4π) indicate the wavelengths to form the resonant cavity modes in the WS_2_ flakes.

**Figure 6 nanomaterials-12-02388-f006:**
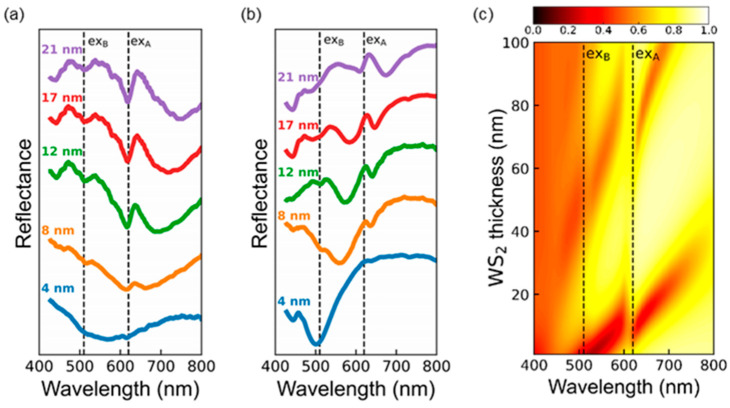
Measured optical reflectance spectra of WS_2_ flakes on SiO_2_/Si (**a**) before and (**b**) after deposition of 30-nm-thick Au thin films. (**c**) TMM-calculated reflectance spectra of Au(30 nm)-coated WS_2_ flakes on SiO_2_/Si. The exciton resonance (ex_A_ and ex_B_) wavelengths of WS_2_ are denoted as dashed lines in (**a**–**c**).

## Data Availability

The data presented in this study are available on request from the corresponding author.

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
