# Peer review of "Self-Hybridized Exciton-Polaritons in Sub-10-nm-Thick WS2 Flakes: Roles of Optical Phase Shifts at WS2/Au Interfaces"

_nanomaterials, 2022, doi:10.3390/nano12142388_

Round 1

Reviewer 1 Report

The referee of this work tried during the reading of the paper to find arguments which would eliminate any plasmonic effect in the spectra of reflectance measured without having found a convincing way. Could the authors offer a small discussion on this point?

Reviewer 2 Report

See attached 

Reviewer 3 Report

This study appears to have been conducted carefully and is well-presented. However, I do have a few questions and comments.

1. Could the authors please provide more information about the parameters adopted for the FDTD study. I assume that FDTD is used in order to model the effects of nonlinear (excitonic) absorption, is that correct? From which reference is the nonlinear absorption data obtained? How do you account for the intensity dependence of this effect? If you are not modelling non-linear absorption, then its not clear why a simple TMM model wouldn't give you the same electric field amplitude and phase data.

2. Figures 2 were obtained using materials data from reference 28. However, reference 28 gives data for WS2 as a free-standing film or on fused silica, but not on gold. How where the parameters for WS2 on gold obtained?

3, The abstract has a sentence that could be improved for readability: 'The self-hybridized EP modes should depend on the TMD thickness, which directly determines the resonance wavelength.'. What does 'should' mean in this context? It might be better to reword this sentence so that it says something like 'The self-hybridized EP modes are expected  to depend on the TMD thickness...'

Round 2

Reviewer 1 Report

Partially convincing answer, which should keep the readers of this paper thinking!

Reviewer 2 Report

Agree to accept